# a1: Steep Test-time Scaling Law via Environment Augmented Generation

## Abstract

Large Language Models (LLMs) have made remarkable breakthroughs in reasoning, yet continue to struggle with hallucinations, logical errors, and inability to self-correct during complex multi-step tasks. Current approaches like chain-of-thought prompting offer limited reasoning capabilities that fail when precise step validation is required. We propose Environment Augmented Generation (EAG), a framework that enhances LLM reasoning through: (1) real-time environmental feedback validating each reasoning step, (2) dynamic branch exploration for investigating alternative solution paths when faced with errors, and (3) experience-based learning from successful reasoning trajectories. Unlike existing methods, EAG enables deliberate backtracking and strategic replanning through tight integration of execution feedback with branching exploration. Our a1-32B model achieves state-of-the-art performance among similar-sized models across all benchmarks, matching larger models like o1 on competition mathematics while outperforming comparable models by up to 24.4 percentage points. Analysis reveals EAG's distinctive scaling pattern: initial token investment in environment interaction yields substantial long-term performance dividends, with advantages amplifying proportionally to task complexity.

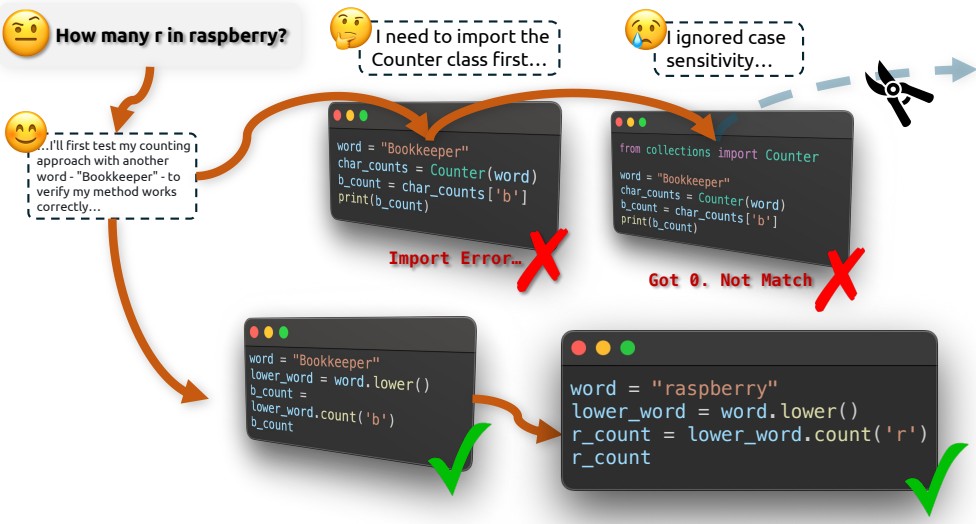

Figure 1: Illustration of the Environment Augmented Generation (EAG) framework solving a character counting task. The model explores multiple solution paths with instant feedback.

## 1 Introduction

Large Language Models (LLMs) have made remarkable breakthroughs in various domains recently (Brown et al., 2020; OpenAI, 2024; DeepSeek-AI et al., 2025b; Qwen et al., 2025), particularly in reasoning capabilities where they can generate intermediate reasoning steps, substantially improving

performance on complex tasks(Kojima et al., 2022b; DeepSeek-AI et al., 2025a; Team, 2024; Team et al., 2025). Despite these advances, reasoning in complex multi-step tasks remains a significant challenge, with models continuing to suffer from hallucinations, logical errors, and an inability to self-correct during extended reasoning chains (Yao et al., 2023a; Schick et al., 2023; Nakano et al., 2021; Carrow et al., 2024; Shao et al., 2024a). However, such models still rely on the model to plan out an entire solution in one forward pass, with no feedback until the final answer is produced. This fundamental limitation means the model's internal plan is unchecked: if an early reasoning step is flawed, the model will continue down a wrong path, often leading to compounding errors or hallucinations (Lightman et al., 2023; Wan et al., 2025; Li et al., 2025c). Fundamentally, static one-pass generation leaves no mechanism to verify intermediate steps or reverse errors, making complex multistep reasoning an open challenge in the field (Huang et al., 2022c;b).

Recent research has explored several promising directions to address these reasoning limitations. External verification approaches leverage tool use and feedback mechanisms (Nakano et al., 2021; Karpas et al., 2022; Yao et al., 2023a; Schick et al., 2023; Das et al., 2024; Wang et al., 2024a; Fourney et al., 2024) to ground responses in factual information. Planning-oriented methods enable LLMs to generate code-form plans (Wen et al., 2024) or explore multiple reasoning paths (Yao et al., 2023b; Hao et al., 2023; Zhang et al., 2025). Tool-integrated reasoning systems (Parisi et al., 2022; Gou et al., 2023; Li et al., 2025a) combine natural language reasoning with computational tools, while self-improvement techniques use refinement (Zelikman et al., 2022; Huang et al., 2022a) and reflection (Shinn et al., 2023; Madaan et al.,

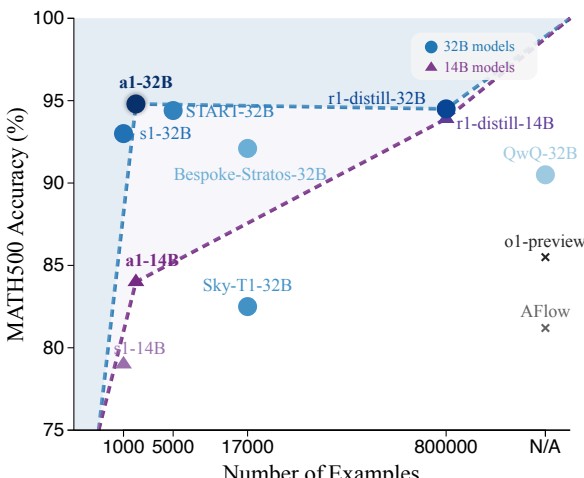

Figure 2: Model performance on MATH500 benchmark versus training data size. Dashed lines show scaling trends for a1. Our a1-32B achieves superior performance with fewer training examples compared to baseline models.

2023; Li et al., 2025a) to enhance reasoning quality. Despite these advances, key limitations persist: tool-using agents typically follow linear reasoning paths (Qin et al., 2023; Li et al., 2025b), planning methods lack real-time verification of steps, and exploratory approaches rarely integrate feedback with dynamic replanning (Zhang et al., 2025). These gaps indicate the need for a framework that unifies immediate verification, branching exploration, and adaptive learning.

In this work, we propose Environment Augmented Generation (EAG) to fill this gap. EAG is a new paradigm for LLM reasoning that tightly couples the model with an external environment during the generation process, transforming reasoning into an interactive, feedback-driven loop. EAG introduces three key innovations: (1) Real-Time Environmental Feedback: At **each** step of reasoning, the model queries an external environment (such as a computational engine, knowledge base, or simulator) to validate the step or obtain new information before proceeding. This immediate feedback acts as a guardrail, catching hallucinations or logical errors on the fly. Instead of only checking a final answer, EAG constantly checks intermediate conclusions – much like a mathematician verifies each line of a proof – greatly mitigating error propagation. (2) Dynamic Branch Exploration: Rather than committing to a single chain-of-thought, EAG explores multiple branches of reasoning in a goal-directed manner. The LLM can maintain several hypothetical solution paths simultaneously, branching when uncertainty is high or multiple approaches seem promising (similar to how one might try different problem-solving strategies). Branches that lead to dead-ends (as indicated by environmental feedback or logical contradiction) can be pruned, and effort focused on fruitful directions. This dynamic search enables strategic lookahead and backtracking, incorporating the strengths of approaches like ToT but augmented with real feedback signals. (3) Trajectory-Based Learning: EAG treats each reasoning attempt as a trajectory through a state space (defined by problem states and reasoning steps). Successful trajectories – those that reach a correct solution with all steps validated – are collected as valuable experiences. The model is then iteratively refined on these

trajectories, via fine-tuning or reinforcement learning, so that it internalizes the effective reasoning patterns. Over time, the LLM improves its policy of reasoning: it learns to avoid invalid steps and favor actions that led to success in the past. This trajectory-based learning paradigm allows the model to learn from its own reasoning experience, continuously closing the loop between planning and feedback.

| Method | Environmental Interaction | | Learning Efficiency | | Expressiveness | | |
| --- | --- | --- | --- | --- | --- | --- | --- |
| | Integrated | Planning | Data | Parameters | Structuring | Versatility | Interpretability |
| CoT (Wei et al., 2023) | ✗ | ✗ | ✗ | N/A | ✗ | ✓ | ✓ |
| AFLOW (Zhang et al., 2025) | ✗ | ✓ | ✗ | N/A | ✗ | ✓ | ✗ |
| o1-like foundation models | ✗ | ✗ | ✗ | ✗ | ✗ | ✗ | ✓ |
| CODEPLAN(Wen et al., 2024) | ✓ | ✓ | ✗ | ✓ | ✓ | ✓ | ✓ |
| s1 (Muennighoff et al., 2025) | ✗ | ✗ | ✗ | ✓ | ✗ | ✗ | ✓ |
| START (Li et al., 2025a) | ✓ | ✗ | ⊁ | ✓ | ✓ | ✓ | ✓ |
| OURS | ✓ | ✓ | ✓ | ✓ | ✓ | ✓ | ✓ |

Table 1: Performance metrics of different reasoning methods across tool use, learning capabilities, and expressiveness dimensions.

By combining these three components, EAG offers a theoretically grounded and practically powerful framework for LLM reasoning. It departs from prior single-pass or dual-pass methods, instead viewing reasoning as an interactive decision-making process akin to an agent navigating a search problem with guidance. In effect, EAG transforms the LLM into a planner that can observe consequences (via the environment), explore alternatives, and learn from trials. This is a paradigm shift: the classical view of prompting LLMs with a static prompt is replaced by a feedback-driven loop that more closely resembles how humans solve problems (trying steps, checking results, revising plans). We hypothesize and will demonstrate that EAG yields more reliable, accurate, and interpretable reasoning. Theoretically, EAG aligns generation with an external verification signal, which can be analyzed in terms of search algorithms and reinforcement learning, providing a new lens to study LLM reasoning. Practically, EAG can solve multi-step tasks that were previously intractable for LLMs alone, and it continually improves with more experience.

## 2 RELATED WORK

**Reasoning via Prompting and Multi-path Exploration.** Chain-of-thought prompting (Wei et al., 2023) pioneered multi-step reasoning in LLMs, leading to advanced techniques (Press et al., 2023; Imani et al., 2023; Hong et al., 2024). Recent work explores multi-path exploration (OpenAI, 2024) and test-time scaling (Muennighoff et al., 2025). State-of-the-art models combine these approaches with SFT or RL (Team, 2024; DeepSeek-AI et al., 2025a; InternLM Team, 2023; Team et al., 2025), while distillation extends benefits to smaller models (Huggingface, 2025; Qin et al., 2024; Ye et al., 2025). Tree-based exploration (Yao et al., 2023b) and iterative refinement (Shinn et al., 2023) provide complementary capabilities.

**Domain-Specific Reasoning and Tool Integration.** Specialized training has enhanced LLM capabilities in mathematics (Yu et al., 2023; Mitra et al., 2024; Shao et al., 2024a), code (Le et al., 2022; Shen et al., 2023), and instruction-following (Cui et al., 2023). Tool integration addresses limitations through calculators (Schick et al., 2023), retrievers (Asai et al., 2024), and code interpreters (Gao et al., 2023). Code execution enhances reasoning via prompting (Gao et al., 2023; Ye et al., 2023; Chen et al., 2023a) or fine-tuning (Gou et al., 2023; Liao et al., 2024; Li et al., 2024a), while code pre-training improves mathematical abilities (Shao et al., 2024b).

**Structured Planning and Decision-Making.** Code structures formalize reasoning across various domains (Madaan et al., 2022; Wang et al., 2022; 2024b). Planning research employs prompting (Wang et al., 2023; Khot et al., 2022) or fine-tuning (Yin et al., 2024; Guan et al., 2024) for plan generation, while recent work explores implicit planning (Zelikman et al., 2024; Cornille et al., 2024).

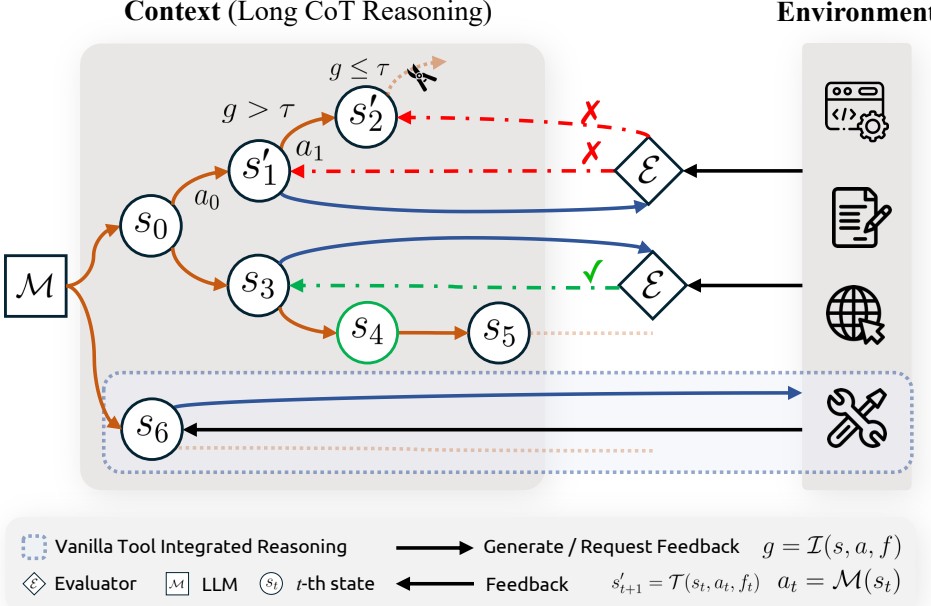

Figure 3: EAG framework. Left: branched state transition graph showing model navigation through states $(s_0, s_1, \ldots)$ with information gain-guided decisions $(g > \tau)$. Right: environmental interfaces providing real-time feedback $(\mathcal{E})$ for step validation. Green checkmarks and red crosses indicate successful and failed paths respectively.

## 3 METHOD

EAG framework formalizes reasoning as a Markov Decision Process (MDP) $(\mathcal{S}, \mathcal{A}, \mathcal{F}, \mathcal{T}, \mathcal{R})$, where $\mathcal{S}$ represents the state space of problem representations and validated reasoning steps, $\mathcal{A}$ denotes the set of possible reasoning actions, $\mathcal{F}$ captures structured environmental feedback, $\mathcal{T}$ defines state transitions, and $\mathcal{R}$ implicitly determines terminal states. The objective is to maximize the trajectory success rate:

$$\max_{\pi} \mathbb{E}_{\pi} \left[ \mathbb{I}(s_T \in \mathcal{S}_{\text{terminal}}) \right] \tag{1}$$

where policy $\pi(a|s)$ is parameterized by the language model. Terminal states $\mathcal{S}_{\text{terminal}}$ are determined implicitly by the language model generating reasoning termination tokens or through environment feedback indicating problem resolution.

### 3.1 STRUCTURED FEEDBACK AND BRANCH EXPLORATION

We introduce a structured feedback representation $f = (v, \sigma, \delta)$ where $v \in \mathbb{R} \cup \{\emptyset\}$ represents numerical values or error codes, $\sigma \in \Sigma$ denotes semantic type information, and $\delta \in \mathcal{D}$ captures descriptive content. This enables rich information transfer between the environment and model.

### 3.2 DYNAMIC BRANCH EXPLORATION MECHANISM

We define a branch value function $V_B(s)$ that combines information gain, path progress, and cost constraints:

$$V_B(s) = \underbrace{\lambda_I D_{\text{KL}} \left( P(f|a, s) \| P_{\text{prior}}(f) \right)}_{\text{information gain}} + \underbrace{\lambda_P \frac{t}{T} \cdot \mathbb{I}[\text{Success}(f)]}_{\text{path progress}} + \underbrace{\lambda_C \mathbb{I}[f \text{ contains errors}]}_{\text{cost constraint}} \tag{2}$$

where $P_{\text{prior}}(f)$ represents a baseline distribution over expected feedback types estimated from historical reasoning trajectories, serving as a reference point for measuring the information value

of new feedback. For practical implementation, we decompose the information gain into weighted components:

$$I(s, a, f) = w_v V(f) + w_e E(f) + w_p P(a, f) \tag{3}$$

where $V(f)$, $E(f)$, and $P(a, f)$ respectively evaluate value information, error information, and progress.

### 3.3 FEEDBACK-GUIDED ACTION SELECTION

The model generates subsequent reasoning steps using a hybrid policy that combines language model predictions with feedback guidance:

$$\pi_{\text{hybrid}}(a|s) = \alpha \cdot \pi_{\text{LM}}(a|s) + (1 - \alpha) \cdot \pi_{\text{feedback}}(a|s, f_{<t}) \tag{4}$$

where $\pi_{\text{feedback}}$ is implemented through an attention mechanism:

$$\pi_{\text{feedback}} = \text{softmax}\left(W \cdot [h_{\text{LM}}; h_{\text{feedback}}]\right) \tag{5}$$

Here, $h_{\text{LM}}$ is the language model's final layer hidden state. The feedback representation $h_{\text{feedback}}$ is derived via a feedback encoder processing the structured feedback components $(v, \sigma, \delta)$. This encoder maps feedback to a continuous representation suitable for integration. The mechanism combining $h_{\text{LM}}$ and $h_{\text{feedback}}$ to influence action selection is optimized jointly with the LM through SFT. This allows the model to learn an optimal weighting between its own predictions and feedback-guided corrections. The state transition logic, elaborated in Algorithm 1, is defined by first generating an exploration state (Eq 6) and then committing or replanning based on branch value ($V_B$) and feedback success (Eq 7):

$$s'_{t+1} = s_t \oplus (a_t, f_t) \tag{6}$$

$$s_{t+1} = \begin{cases} s'_{t+1} \oplus (a_{t+1}, f_{t+1}) & \text{if } V_B(s_t) > \tau \text{ and } \texttt{Success}(f_{t+1}) \\ \texttt{Replan}(s_t, f_t) & \text{otherwise} \end{cases} \tag{7}$$

### 3.4 BRANCH EXPLORATION

Algorithm A.1 presents our Branch Exploration (BEx) procedure that formalizes the exploration process as a heuristic graph search. BEx maintains a set of active branches $B$ and iteratively expands promising paths while pruning those that fail to yield progress:

1. **Branch Set Initialization**: $B_0 = \{s_0\}$
2. **Depth-First Expansion**: For each depth $d \leq D_{\max}$:

$$B_{d+1} = \bigcup_{s \in B_d} \{\mathcal{T}(s, a, f) \mid a \sim \pi(\cdot|s), f = \mathcal{E}(a), V_B(s') \geq \tau\} \tag{8}$$

3. **Pruning Strategy**: Remove branches where $V_B(s) < \tau$ or $C(s) > C_{\max}$, where $\tau$ is a configurable information gain threshold determining whether a branch is promising enough to continue exploring
4. **Terminal State Detection**: If $\exists s \in B_d$ satisfying $s \in \mathcal{S}_{\text{terminal}}$, return the corresponding solution

### 3.5 ALIGNMENT BETWEEN MDP FORMALISM AND SUPERVISED LEARNING

We adapt the MDP formalism for reasoning, diverging from standard reinforcement learning. Instead of direct policy optimization, the MDP guides:

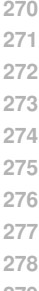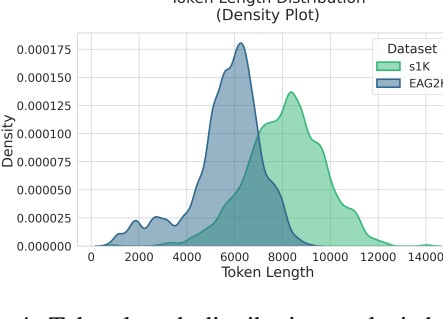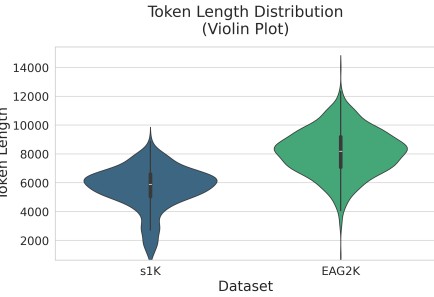

Figure 4: Token length distribution analysis between s1K and EAG2K datasets. The violin plots (right) show the overall distribution shapes and ranges, with EAG2K exhibiting a higher median length and wider spread. The density plots (left) highlight the shift towards longer sequences in EAG2K, with peaks at approximately 6000 and 8000 tokens for s1K and EAG2K respectively.

1. **Trajectory Collection**: The datasets only contains successful reasoning trajectories:

$$\mathcal{D}_{\text{EAG}} = \{(s_t, a_t, f_t, s_{t+1})_{t=0}^T | s_T \in \mathcal{S}_{\text{terminal}}\} \tag{9}$$

2. **Supervised Learning Objective**: We train the language model through supervised fine-tuning (SFT) to maximize the conditional likelihood of actions given states:

$$\mathcal{L}_{\text{SFT}} = -\mathbb{E}_{(s_t, a_t) \sim \mathcal{D}_{\text{EAG}}} \left[ \log \pi_\theta(a_t|s_t) \right] \tag{10}$$

This bypasses RL's exploration issues by directly learning from diverse, verified reasoning examples. The resulting model efficiently aligns with MDP principles and implicitly internalizes feedback/exploration, exhibiting emergent reasoning without explicit value functions.

## 4 DATASET

The Environment Augmented Generation (EAG) framework requires reasoning trajectories that integrate real-time environmental feedback. To enable this capability, we construct the **EAG-2K** dataset, a curated collection of 2,000 interactive reasoning traces derived from the s1 dataset. Our dataset transformation process emphasizes three critical objectives: (1) preserving the model's intrinsic reasoning ability, (2) simulating code-environment interactions with structured feedback, and (3) balancing trajectory length and computational feasibility. Below, we detail the construction methodology, data composition, and quality control mechanisms.

### 4.1 DATA TRANSFORMATION FRAMEWORK

We transform the s1 dataset (1,000 reasoning traces across mathematical, scientific, and coding domains) by converting natural language reasoning into executable Python code with environmental feedback. Using few-shot prompting with `claude-3.7-sonnet`, we create code blocks for computations, validations, and simulations, marked with `<|execute|>` tags. Executions in a Python sandbox generate structured feedback (value, type, status) enclosed in `<|feedback|>` tags. Our transformation targets the LongCoT portion between `<|im_start|>think` and `<|im_start|>answer` tags—where step-by-step calculations and logical deductions occur—making it ideal for validating each reasoning step with executable code and feedback. To expand from the original 1,000 s1 traces to our 2,000-sample EAG-2K dataset, we augment complex reasoning cases with multiple solution paths and error-correction trajectories, effectively doubling the dataset size while enriching it with branch exploration examples.

### 4.2 DATA COMPOSITION

The dataset is partitioned into three subsets to balance capability retention and interactive learning:

**Raw Subset (200 samples).** To preserve the model's inherent reasoning ability, 10% of the original s1 trajectories remain unmodified. These samples are selected based on two criteria: *diversity* (covering

mathematics, code debugging, and scientific QA) and *difficulty* (problems where Qwen2.5-32B achieves <30% accuracy without environmental feedback). This subset ensures the model retains baseline problem-solving strategies independent of external tools.

| Metric | Initial | Retry@1 | Retry@2 | Retry@3 |
|---|---|---|---|---|
| Avg. Tokens | 6,244 | 7,881 | 9,900 | 15,000 |
| Success Rate | 62% | 89% | 95% | 98% |

Table 2: Trajectory statistics for Iterative-Refinement process, showing token length and success rate changes across retry attempts.

**Iterative-Refinement Subset (800 samples).** This subset captures dynamic error recovery patterns by preserving trajectories where environmental feedback triggers immediate code regeneration. Samples are included if they demonstrate: (1) failed initial executions with recoverable errors (e.g., type mismatches or missing dependencies), and (2) feedback-driven code revisions within three attempts. Each revision cycle follows the pattern:

---

**Error Correction with Environment Feedback**

```
<|execute|>
x = 5 / 0   # Initial error
<|execute_end|>
<|feedback|>
ZeroDivisionError: division by zero
<|feedback_end|>
Oops!  I've encountered a ZeroDivisionError.  I'm trying to divide
5 by zero...
I should check if the denominator is zero before dividing...  Let
me fix this by checking if y is zero before dividing by it.
<|execute|>
x = 5 / (y if y != 0 else 1)   # Revised code using feedback
<|execute_end|>
```

Figure 5: Example of an iterative refinement cycle with execution, feedback, and correction.

**Direct-Execution Subset (1,000 samples).** Promotes efficient environment-coupled reasoning by enforcing single-attempt code execution, effectively using only the successful version. Trajectories over 16K tokens are shortened by isolating core computations and retaining only this final successful code. This trains the model to prioritize correct implementations over error exploration, particularly effective for formulaic problems where extensive debugging offers little value.

## 5 EXPERIMENTS

### 5.1 SETUP

We perform supervised finetuning on Qwen2.5-32B-Instruct using our EAG-2K dataset to obtain the a1-32B model with environment augmented reasoning capabilities. Finetuning took approximately 12 hours on 8 NVIDIA A100 GPUs with PyTorch FSDP. For more details, please refer to Appendix A.5.

### 5.2 RESULTS

Table 3 validates EAG's effectiveness: our a1-32B model achieves state-of-the-art performance among 32B models across all evaluated reasoning benchmarks. Its notable 74.4% on AIME24 matches the much larger o1 model and significantly outperforms peers like QwQ-32B-Preview (+24.4%) and s1-32B (+17.7%). This strong, consistent performance extends to AIME25 (50.0%), MATH500 (94.8%), and GPQA (63.4%), highlighting EAG's general applicability, likely stemming from its structured environmental feedback mechanism. Furthermore, matching large models like o1 (>100B parameters) demonstrates significant parameter efficiency, positioning EAG as an effective,

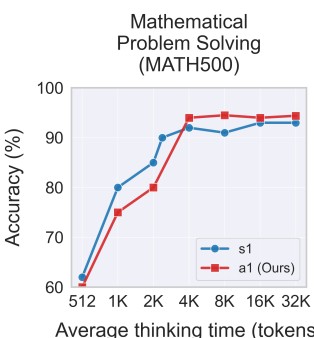 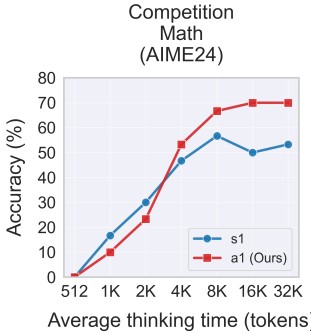 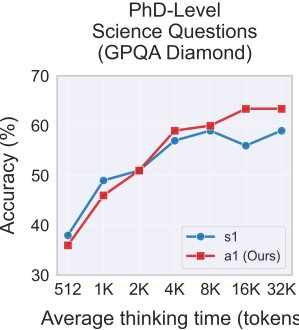

Figure 6: Performance comparison between our `a1` model and baseline `s1` across different thinking time budgets. For MATH500, `a1` shows stronger performance at higher token counts despite starting lower. In more challenging domains like AIME24 and GPQA Diamond, the advantage of `a1` becomes more pronounced with increased thinking time, demonstrating superior scaling properties of our environment augmented approach.

complementary approach to model scaling for reasoning, offering a potentially favorable reasoning-computation trade-off.

| Method | GPQA | MATH500 | AIME24 | AIME25 |
|---|---|---|---|---|
| Qwen2.5-32B | 46.4 | 75.8 | 23.3 | 13.3 |
| Qwen2.5-Coder-32B | 33.8 | 71.2 | 20.0 | - |
| Llama3.3-70B | 43.4 | 70.8 | 36.7 | - |
| GPT-4o[†] | 50.6 | 60.3 | 9.3 | - |
| o1-preview[†] | 75.2 | 85.5 | 44.6 | 37.5 |
| o1[†] | 77.3 | 94.8 | 74.4 | - |
| DeepSeek-R1-Distill-Qwen-32B[†] | 62.1 | 94.3 | 72.6 | 46.7 |
| s1-32B[†] | 59.6 | 93.0 | 50.0 | 33.3 |
| Search-o1-32B[†] | **63.6** | 86.4 | 56.7 | - |
| QwQ-32B-Preview | 58.1 | 90.6 | 50.0 | 36.7 |
| START | 63.6 | 94.4 | 66.7 | 47.1 |
| **a1-32B (Ours)** | 63.4 | **94.8** | **74.4** | **50.0** |

Table 3: Main results on reasoning tasks. We report Pass@1 metric. Best results for 32B models are in **bold**. Larger/non-proprietary models shown in gray. Symbol "†" indicates the results are from their official releases.

Figure 6 illustrates the scaling advantages of our `a1` model compared to baseline `s1` when provided with increased token budgets across three benchmark domains. The analysis reveals EAG's characteristic **steep scaling** pattern. Initially, `a1` may lag `s1` at low token budgets (e.g., 512-2K on MATH500). This is due to the token overhead required for environment interaction via `<|execute|>`/`<|feedback|>` cycles. However, this initial investment yields significant long-term dividends. A distinct inflection point typically emerges (around 4K-8K tokens), after which `a1`'s performance rapidly surpasses the baseline and its advantage accelerates. This steep improvement is particularly pronounced in complex domains like AIME24 (achieving a 15 pp advantage at 32K tokens) and GPQA Diamond (dominating beyond 4K tokens). This behavior validates our framework's emphasis: the cost of incremental feedback is quickly outweighed by the benefit of

empirically validated, higher-information-density reasoning paths, an advantage that amplifies with task complexity.

## 5.3 ABLATION STUDY

| Model Variant | AIME24 | MATH500 | GPQA |
|---|---|---|---|
| s1-32B (baseline) | 50.0 | 93.0 | 59.6 |
| a1-32B w/o B.E. | 53.3 | 90.0 | 61.6 |
| a1-32B with num. | 56.7 | 93.4 | 62.3 |
| a1-32B (full) | **74.4** | **94.8** | **63.4** |

Table 4: Ablation study. "w/o B.E." removes dynamic branch exploration, while "with num. only" restricts the feedback to numerical values only, removing error descriptions and semantic type information.

Our ablation study reveals important insights about the contribution of each component in our EAG framework. Removing dynamic branch exploration ("w/o B.E.") severely impacts performance on complex reasoning tasks like AIME24, where accuracy drops by 21.1% points. This suggests that the ability to explore alternative solution paths when faced with errors is crucial for solving challenging mathematical problems that require precise step validation. Similarly, restricting the model to numerical feedback only without error descriptions or semantic type information ("with num.") results in a substantial performance drop, particularly on AIME24 (17.7%). This demonstrates the importance of rich, structured feedback in guiding the reasoning process. The full EAG implementation consistently outperforms all ablated versions across all benchmarks, confirming our hypothesis that the integration of both components—dynamic branch exploration and rich structured feedback—is essential for maximizing reasoning capabilities in complex multi-step tasks.

## 6 CONCLUSION

This paper introduces Environment Augmented Generation (EAG), a framework that transforms how language models approach complex reasoning tasks through real-time environmental feedback and dynamic branch exploration. Our empirical results demonstrate significant improvements: our a1-32B model achieves state-of-the-art performance among similar-sized models across all benchmarks, matching larger models like o1 on competition mathematics. The success of EAG reveals a distinctive scaling pattern: initial token investment in environment interaction yields substantial long-term performance dividends, with advantages amplifying proportionally to task complexity. EAG's theoretical framework demonstrates how environment interactivity and systematic branch exploration together establish a new paradigm for reliable machine reasoning, particularly for problems requiring precise multi-step calculation and logical verification. Beyond immediate performance gains, EAG's approach suggests a fundamental shift from static generation to interactive reasoning processes, opening new avenues for developing more reliable and verifiable AI systems. The framework's ability to achieve comparable performance to much larger models while maintaining parameter efficiency indicates promising directions for democratizing advanced reasoning capabilities across resource-constrained environments.

## ETHICS STATEMENT

This research enhances LLM reasoning capabilities through the Environment Augmented Generation (EAG) framework, aiming to develop more verifiable and accurate AI reasoning systems. While our EAG-2K dataset, derived from s1, simulates code execution in a controlled environment, we acknowledge potential limitations from simulated feedback and model-generated data. Though improving reasoning reliability advances trustworthy AI, we recognize the dual-use potential of enhanced reasoning capabilities. The EAG framework's computational demands during inference raise considerations about energy consumption and resource accessibility. However, we believe this trade-off between initial computational cost and improved performance is justified in pursuing more robust and verifiable AI systems that prioritize safety and reliability.

## REPRODUCIBILITY STATEMENT

We took concrete steps to make our results easy to replicate. Dataset sources, preprocessing, model variants, and training/evaluation protocols are specified in Sections 3–5, with full hyperparameters, prompts, seeds, and environment versions in Appendices B–D. An anonymized repository in the supplementary materials includes code, configs, and scripts to reproduce all reported tables/figures from a clean checkout. Together, these materials enable reliable re-runs and straightforward extensions.

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

# A APPENDIX

## A.1 BRANCH EXPLORATION ALGORITHM

---

**Algorithm 1** Branch Exploration

---

**Require:** Problem $p$, Language model $\mathcal{M}$, Environment $\mathcal{E}$, Threshold $\tau$
**Ensure:** Solution trajectory $\tau = (s_0, a_0, f_0, ..., s_T)$ where $s_T \in \mathcal{S}_{\text{terminal}}$
1:  $s_t \leftarrow (p, \emptyset, \emptyset)$                $\triangleright$ Initialize state with problem and empty histories
2:  **while** $s_t \notin \mathcal{S}_{\text{terminal}}$ **do**
3:      $a_t \leftarrow \mathcal{M}(s_t), f_t \leftarrow \mathcal{E}(a_t)$           $\triangleright$ Generate action and get feedback
4:      **if** $\text{Success}(f_t)$ **then**
5:          $s_t \leftarrow s_t \oplus (a_t, f_t)$             $\triangleright$ Extend current stable state
6:      **else**
7:          $(g, p, c) \leftarrow (I(s_t, a_t, f_t), P(s_t), C(s_t))$      $\triangleright$ Compute branch metrics
8:          $V_B(s_t) \leftarrow \lambda_I g + \lambda_P p + \lambda_C c$        $\triangleright$ Calculate branch value
9:          **if** $V_B(s_t) > \tau$ **then**
10:           $s'_{t+1} \leftarrow s_t \oplus (a_t, f_t)$         $\triangleright$ Create exploration state
11:           $a_{t+1} \leftarrow \mathcal{M}(s'_{t+1}), f_{t+1} \leftarrow \mathcal{E}(a_{t+1})$        $\triangleright$ Try repair
12:           $s_t \leftarrow \text{Success}(f_{t+1})? s'_{t+1} \oplus (a_{t+1}, f_{t+1}) : \text{Replan}(s_t, f_t)$
13:          **else**
14:           $s_t \leftarrow \text{Replan}(s_t, f_t)$          $\triangleright$ Abandon current branch
15:          **end if**
16:      **end if**
17: **end while**
18: **return** $(s_0, a_0, f_0, ..., s_T)$

---

## A.2 DATASET EXAMPLE

Below is a sample from our EAG-2K dataset showing how environmental feedback guides mathematical reasoning:

---

**Dataset Example: Mathematical Problem Solving with EAG**

```
<|im_start|>system
You are Qwen, created by Alibaba Cloud. You are a helpful assistant
    .<|im_end|>
<|im_start|>user
Given a rational number, write it as a fraction in lowest terms and
    calculate the product of the resulting numerator and
    denominator. For how many rational numbers between 0 and 1 will
    $20_{}^{}!$ be the resulting product?<|im_end|>
<|im_start|>assistant
<|im_start|>think
Let the rational number be $q$. Since $0 < q < 1$, we can write $q
    = \frac{a}{b}$ where $a, b$ are positive integers and $a < b$.
The problem states that the fraction is written in lowest terms.
    This means $\gcd(a, b) = 1$.
The problem also states that the product of the resulting numerator
    and denominator is $20!$. So, $a \times b = 20!$.

We are looking for the number of pairs of positive integers $(a, b)
    $ such that:
1.  $a \times b = 20!$
2.  $\gcd(a, b) = 1$
3.  $a < b$
```

---

Let the prime factorization of $20!$ be $20! = p_1^{e_1} p_2^{e_2} \cdots p_k^{e_k}$, where $p_1, p_2, \ldots, p_k$ are distinct prime numbers and $e_1, e_2, \ldots, e_k$ are their exponents.
Since $a \times b = 20!$ and $\gcd(a, b) = 1$, it means that for each prime factor $p_i$ of $20!$, the entire power $p_i^{e_i}$ must belong either to $a$ or to $b$. It cannot be split between them, otherwise $p_i$ would divide both $a$ and $b$, contradicting $\gcd(a, b) = 1$.

So, to form a pair $(a, b)$ satisfying conditions 1 and 2, we need to decide for each distinct prime factor $p_i$ whether its corresponding prime power $p_i^{e_i}$ goes into the factorization of $a$ or $b$.
First, let's find the distinct prime factors of $20!$. These are the prime numbers less than or equal to 20.

```
<|execute|>
def is_prime(n):
    if n < 2:
        return False
        for i in range(2, int(n**0.5) + 1):
            if n % i == 0:
                return False
        return True

distinct_primes_in_20_factorial = [p for p in range(2, 21) if
    is_prime(p)]
print(distinct_primes_in_20_factorial)
k = len(distinct_primes_in_20_factorial)
print(f"Number of distinct prime factors (k): {k}")
<|execute_end|>
<|feedback|>
[2, 3, 5, 7, 11, 13, 17, 19]
Number of distinct prime factors (k): 8
<|feedback_end|>
```

There are $k=8$ distinct prime factors in $20!$. Let these primes be $P=\{2,3,5,7,11,13,17,19\}$.
For each prime $p \in P$, let $p^e$ be the highest power of $p$ dividing $20!$.
A pair $(a,b)$ such that $a \times b = 20!$ and $\gcd(a,b)=1$ is formed by choosing a subset $S \subseteq P$.
Then $a = \prod_{p \in S} p^e$ and $b = \prod_{p \in P \setminus S} p^e$.
The number of subsets $S$ of $P$ is $2^k = 2^8 = 256$.
Each subset $S$ corresponds to a unique ordered pair $(a,b)$ satisfying conditions 1 and 2.

Now we need to consider the condition $a < b$.
Since $a \times b = 20!$, the condition $a < b$ is equivalent to $a < \sqrt{20!}$.
Can $a = b$? This would imply $a^2 = 20!$, meaning $20!$ is a perfect square.
A factorial $N!$ for $N > 1$ is never a perfect square. This is because, by Bertrand's Postulate, there is always a prime $p$ such that $\lfloor N/2 \rfloor < p \leq N$. The exponent of this prime $p$ in the prime factorization of $N!$ is $\lfloor N/p \rfloor + \lfloor N/p^2 \rfloor + \cdots = 1 + 0 + \cdots = 1$. Since the exponent is odd, $N!$ cannot be a perfect square.
For $N = 20$, the largest prime is 19. $\lfloor 20/19 \rfloor = 1$. So the exponent of 19 in $20!$ is 1. Thus $20!$ is not a perfect square, and $a \neq b$.

```
Since $a \neq b$, for every pair $(a,b)$ satisfying conditions 1
    and 2, either $a < b$ or $b < a$.
The total number of ordered pairs $(a,b)$ satisfying conditions 1
    and 2 is $2^k = 2^8 = 256$.
These pairs can be grouped into pairs of the form $\{(a,b), (b,a)\}
    $. Since $a \neq b$, each group has exactly two distinct pairs.
One pair in the group satisfies $a < b$, and the other satisfies $b
     < a$.
Therefore, exactly half of the pairs satisfy $a < b$.
The number of pairs $(a,b)$ satisfying conditions 1, 2, and 3 is $\
    frac{1}{2} \times 2^k = 2^{k-1}$.
With $k = 8$, the number of pairs is $2^{8-1} = 2^7$.
<|execute|>
k = 8
result = 2**(k - 1)
print(result)
<|execute_end|>
<|feedback|>
128
<|feedback_end|>

So, there are 128 pairs $(a,b)$ that satisfy all three conditions.
    Each such pair corresponds to a unique rational number $q = a/b$
     between 0 and 1, in lowest terms, such that the product of the
    numerator and denominator is $20!$.

Let's double check with a smaller example, say 6!.

$6! = 720 = 2^4 \times 3^2 \times 5^1$.
The distinct prime factors are $P = \{2,3,5\}$, so $k = 3$.
The number of pairs $(a,b)$ with $a \times b = 720$, $\gcd(a,b) = 1
    $, and $a < b$ should be $2^{k-1}$.
<|execute|>
k_check = 3
result_check = 2**(k_check - 1)
print(result_check)
<|execute_end|>
<|feedback|>
4
<|feedback_end|>
Let's list the pairs $(a,b)$ satisfying $a \times b = 720$ and $\
    gcd(a,b) = 1$.
The subsets of $P$ are $\emptyset, \{2\}, \{3\}, \{5\}, \{2,3\},
    \{2,5\}, \{3,5\}, \{2,3,5\}$.
Corresponding pairs $(a,b)$:

$S = \emptyset$: $a = 1$, $b = 2^4 \times 3^2 \times 5^1 = 720$.
    Pair (1, 720). $a < b$.

$S = \{2\}$: $a = 2^4 = 16$, $b = 3^2 \times 5^1 = 45$. Pair (16,
    45). $a < b$.

$S = \{3\}$: $a = 3^2 = 9$, $b = 2^4 \times 5^1 = 80$. Pair (9, 80)
    . $a < b$.

$S = \{5\}$: $a = 5^1 = 5$, $b = 2^4 \times 3^2 = 144$. Pair (5,
    144). $a < b$.

$S = \{2,3\}$: $a = 2^4 \times 3^2 = 144$, $b = 5^1 = 5$. Pair
    (144, 5). $a > b$.

$S = \{2,5\}$: $a = 2^4 \times 5^1 = 80$, $b = 3^2 = 9$. Pair (80,
    9). $a > b$.
```

$S = \{3,5\}$: $a = 3^2 \times 5^1 = 45$, $b = 2^4 = 16$. Pair (45, 16). $a > b$.

$S = \{2,3,5\}$: $a = 2^4 \times 3^2 \times 5^1 = 720$, $b = 1$. Pair (720, 1). $a > b$.
There are $2^3 = 8$ pairs in total.
The pairs satisfying $a < b$ are (1, 720), (16, 45), (9, 80), (5, 144). There are 4 such pairs.
This matches the formula calculation $2^{k-1} = 4$.

The logic seems correct. For $20!$, $k = 8$, so the number of rational numbers is $128$.<|im_start|>answer
Let the rational number be $q$. We are given that $0 < q < 1$.
When $q$ is written as a fraction in lowest terms, let it be $q = \frac{a}{b}$.
Being in lowest terms means that $a$ and $b$ are positive integers with $\gcd(a,b) = 1$.
The condition $0 < q < 1$ means $0 < \frac{a}{b} < 1$, which implies $a < b$ (since $a, b$ are positive).
We are given that the product of the numerator and denominator is $20!$. So, $a \times b = 20!$.

We need to find the number of pairs of positive integers $(a,b)$ such that:

$a \times b = 20!$

$\gcd(a,b) = 1$

$a < b$

Let the prime factorization of $20!$ be $20! = p_1^{e_1} p_2^{e_2} \cdots p_k^{e_k}$, where $p_1, \ldots, p_k$ are distinct prime numbers and $e_i \geq 1$ are their exponents.
The condition $\gcd(a,b) = 1$ means that $a$ and $b$ share no common prime factors.
Since $a \times b = 20!$, this implies that for each prime factor $p_i$ of $20!$, the entire prime power $p_i^{e_i}$ must be a factor of either $a$ or $b$, but not both.

First, identify the distinct prime factors of $20!$. These are the prime numbers less than or equal to 20.
<|execute|>
```python
def is_prime(n):
    if n < 2:
        return False
    for i in range(2, int(n**0.5) + 1):
        if n % i == 0:
            return False
    return True

distinct_primes_in_20_factorial = [p for p in range(2, 21) if is_prime(p)]
#print(distinct_primes_in_20_factorial)
k = len(distinct_primes_in_20_factorial)
#print(f"Number of distinct prime factors (k): {k}")
```
<|execute_end|>
<|feedback|>
[2, 3, 5, 7, 11, 13, 17, 19]
Number of distinct prime factors (k): 8
<|feedback_end|>

```
There are $k = 8$ distinct prime factors. Let $P = \{2, 3, 5, 7,
    11, 13, 17, 19\}$ be the set of these $k$ distinct prime factors
    .

To form a pair $(a,b)$ satisfying $a \times b = 20!$ and $\gcd(a,b)
     = 1$, we must partition the set of prime powers corresponding
    to primes in $P$ into two sets, one whose product forms $a$, and
     the other whose product forms $b$. This is equivalent to
    choosing a subset $S \subseteq P$.
Let $a = \prod_{p_i \in S} p_i^{e_i}$ and $b = \prod_{p_j \in P \
    setminus S} p_j^{e_j}$ (where $p_i^{e_i}$ is the highest power
    of $p_i$ dividing $20!$).
The number of possible subsets $S$ of $P$ is $2^k$. Since $k = 8$,
    there are $2^8 = 256$ ways to partition the prime powers,
    resulting in 256 ordered pairs $(a,b)$ such that $a \times b =
    20!$ and $\gcd(a,b) = 1$.

Now we must impose the condition $a < b$.
Consider a pair $(a,b)$ formed this way. Since $a \times b = 20!$,
    $a$ and $b$ cannot be equal, because $20!$ is not a perfect
    square (for example, the prime 19 appears with exponent 1 in the
     factorization of $20!$).
Therefore, for any pair $(a,b)$ satisfying conditions 1 and 2, we
    must have either $a < b$ or $b < a$.
The assignment process generates pairs $(a,b)$. If a subset $S$
    generates $(a,b)$, then its complement $P \setminus S$ generates
     the pair $(b,a)$.
Since $a \neq b$, the pairs $(a,b)$ and $(b,a)$ are distinct.
The 256 pairs can be grouped into $256/2 = 128$ sets of the form $
    \{(a,b), (b,a)\}$.
In each set, exactly one pair satisfies the condition $a < b$.
Therefore, the number of pairs $(a,b)$ satisfying all three
    conditions is exactly half of the total number of pairs
    satisfying conditions 1 and 2.
Number of pairs = $\frac{2^k}{2} = 2^{k-1}$.
<|execute|>
k = 8
result = 2**(k - 1)
#print(result)
<|execute_end|>
<|feedback|>
128
<|feedback_end|>
Since $k = 8$, the number of pairs is $2^{8-1} = 2^7 = 128$.

Each such pair $(a,b)$ corresponds to a unique rational number $q =
     a/b$ satisfying the given conditions.
Thus, there are 128 such rational numbers.

Final Answer: The final answer is $128$
<|im_start|>answer
...
```

This example demonstrates how EAG integrates code execution with mathematical reasoning. The model leverages Python to verify prime factorization, calculate exponents using Legendre's formula, and determine the number of valid rational number representations. Each execution step provides immediate feedback that guides subsequent reasoning, enabling accurate step-by-step verification throughout the solution process.

### A.3 ADDITIONAL DATASET EXAMPLE: MODULAR ARITHMETIC

Here is another example showing how EAG aids in solving complex modular arithmetic problems:

**Dataset Example: Modular Arithmetic with EAG**

```
<|im_start|>system
You are Qwen, created by Alibaba Cloud. You are a helpful assistant
    .<|im_end|>
<|im_start|>user
I need to find the remainder when $9 \times 99 \times 999 \times \
    cdots \times \underbrace{99\cdots9}_{\text{999 9's}}$ is divided
     by $1000$.<|im_end|>
<|im_start|>assistant
<|im_start|>think
Let $P$ be the product $9 \times 99 \times 999 \times \cdots \times
     \underbrace{99\cdots9}_{\text{999 9's}}$. We want to find $P \
    pmod{1000}$.

The terms in the product are of the form $a_n = \underbrace{99\
    cdots9}_{n \text{ 9's}}$. We can write $a_n = 10^n - 1$.
The product is $P = a_1 \times a_2 \times a_3 \times \cdots \times
    a_{999}$.

Let's examine the first few terms modulo 1000.
<|execute|>
a1 = 10**1 - 1
a2 = 10**2 - 1
a3 = 10**3 - 1
mod = 1000

rem_a1 = a1 % mod
rem_a2 = a2 % mod
rem_a3 = a3 % mod

print(f"a1 = {a1}, a1 % {mod} = {rem_a1}")
print(f"a2 = {a2}, a2 % {mod} = {rem_a2}")
print(f"a3 = {a3}, a3 % {mod} = {rem_a3}")
# Verify alternate calculation for a3 mod 1000
print(f"a3 % {mod} using -1: {-1 % mod}")
<|execute_end|>
<|feedback|>
a1 = 9, a1 % 1000 = 9
a2 = 99, a2 % 1000 = 99
a3 = 999, a3 % 1000 = 999
a3 % 1000 using -1: 999
<|feedback_end|>

Now let's check the terms for $n \geq 3$. We need to check $a_n =
    10^n - 1 \pmod{1000}$.
<|execute|>
mod = 1000
n = 4
a4 = 10**n - 1
rem_a4 = a4 % mod
print(f"n=4: 10**{n} % {mod} = {10**n % mod}")
print(f"a4 = {a4}, a4 % {mod} = {rem_a4}")

n = 5
a5 = 10**n - 1
rem_a5 = a5 % mod
print(f"n=5: 10**{n} % {mod} = {10**n % mod}")
print(f"a5 = {a5}, a5 % {mod} = {rem_a5}")

# Check congruence to -1
print(f"For n=4, a4 % {mod} == -1 % {mod}: {rem_a4 == (-1 % mod)}")
print(f"For n=5, a5 % {mod} == -1 % {mod}: {rem_a5 == (-1 % mod)}")
```

```
<|execute_end|>
<|feedback|>
n=4: 10**4 % 1000 = 0
a4 = 9999, a4 % 1000 = 999
n=5: 10**5 % 1000 = 0
a5 = 99999, a5 % 1000 = 999
For n=4, a4 % 1000 == -1 % 1000: True
For n=5, a5 % 1000 == -1 % 1000: True
<|feedback_end|>

As observed, for any $n \geq 3$, $10^n$ is divisible by $1000$, so
    $10^n \equiv 0 \pmod{1000}$.
Therefore, for $n \geq 3$, $a_n = 10^n - 1 \equiv 0 - 1 = -1 \pmod
    {1000}$.

The product $P$ is $a_1 \times a_2 \times a_3 \times \cdots \times
    a_{999}$.
Modulo 1000, this becomes:

$P \equiv rem\_a1 \times rem\_a2 \times (rem\_a3) \times (rem\_a4)
    \times \cdots \times (rem\_a_{999}) \pmod{1000}$

$P \equiv 9 \times 99 \times (-1) \times (-1) \times \cdots \times
    (-1) \pmod{1000}$.

We need to count how many terms are congruent to $-1$. These are
    the terms from $a_3$ to $a_{999}$.
<|execute|>
num_neg_one_terms = 999 - 3 + 1
print(f"Number of terms congruent to -1: {num_neg_one_terms}")
<|execute_end|>
<|feedback|>
Number of terms congruent to -1: 997
<|feedback_end|>

So, $P \equiv 9 \times 99 \times (-1)^{997} \pmod{1000}$.

Now we calculate $(-1)^{997}$.
<|execute|>
exponent = 997
base = -1
result_power = base ** exponent
print(f"(-1)**{exponent} = {result_power}")

# Parity check
is_odd = exponent % 2 != 0
print(f"{exponent} is odd: {is_odd}")
print(f"Result should be -1: {result_power == -1}")
<|execute_end|>
<|feedback|>
(-1)**997 = -1
997 is odd: True
Result should be -1: True
<|feedback_end|>

Since 997 is odd, $(-1)^{997} = -1$.
Substituting this back:

$P \equiv 9 \times 99 \times (-1) \pmod{1000}$.

Now, we perform the final calculation.
<|execute|>
mod = 1000
```

```
term1 = 9
term2 = 99
term3 = -1

intermediate_product = (term1 * term2) % mod
print(f"({term1} * {term2}) % {mod} = {intermediate_product}")

final_product_mod = (intermediate_product * term3) % mod
print(f"({intermediate_product} * {term3}) % {mod} = {
    final_product_mod}")

# Adjust if the result is negative to be in [0, mod-1]
final_remainder = final_product_mod if final_product_mod >= 0 else
    final_product_mod + mod
print(f"Final remainder: {final_remainder}")

# Direct calculation verification
direct_calc = (9 * 99 * (-1)) % 1000
print(f"Direct calculation: {(9 * 99 * (-1))} % {mod} = {
    direct_calc}")
final_remainder_direct = direct_calc if direct_calc >= 0 else
    direct_calc + mod
print(f"Final remainder from direct calc: {final_remainder_direct
    }")
<|execute_end|>
<|feedback|>
(9 * 99) % 1000 = 891
(891 * -1) % 1000 = -891
Final remainder: 109
Direct calculation: -891 % 1000 = 109
Final remainder from direct calc: 109
<|feedback_end|>
```

The calculation shows that $P \equiv 891 \times (-1) \equiv -891 \pmod{1000}$.
The remainder must be non-negative, so we add 1000: $-891 + 1000 = 109$.
The remainder is 109.
<|im_start|>answer
Let the given product be $P$.

$P = 9 \times 99 \times 999 \times \cdots \times \underbrace{99\cdots9}_{\text{999 9's}}$

We want to find the remainder when $P$ is divided by $1000$, which is $P \pmod{1000}$.

Let $a_n = \underbrace{99\cdots9}_{n \text{ 9's}}$. We can write $a_n = 10^n - 1$.
The product is $P = a_1 \times a_2 \times a_3 \times \cdots \times a_{999}$.

We examine the terms modulo 1000.

```
<|execute|>
mod = 1000
a1 = 10**1 - 1
a2 = 10**2 - 1
rem_a1 = a1 % mod
rem_a2 = a2 % mod
print(f"a1 mod {mod} = {rem_a1}")
print(f"a2 mod {mod} = {rem_a2}")

# Check terms for n >= 3
```

```
a3 = 10**3 - 1
rem_a3 = a3 % mod
print(f"a3 mod {mod} = {rem_a3}")

a4 = 10**4 - 1
rem_a4 = a4 % mod
print(f"a4 mod {mod} = {rem_a4}")

# Verify congruence to -1 for n >= 3
print(f"a3 congruent to -1 mod {mod}: {rem_a3 == (-1 % mod)}")
print(f"a4 congruent to -1 mod {mod}: {rem_a4 == (-1 % mod)}")
<|execute_end|>
<|feedback|>
a1 mod 1000 = 9
a2 mod 1000 = 99
a3 mod 1000 = 999
a4 mod 1000 = 999
a3 congruent to -1 mod 1000: True
a4 congruent to -1 mod 1000: True
<|feedback_end|>
```

For $n \geq 3$, $10^n$ is divisible by $1000$, so $10^n \equiv 0 \pmod{1000}$.
Thus, $a_n = 10^n - 1 \equiv 0 - 1 = -1 \pmod{1000}$ for $n \geq 3$.

The product $P$ modulo 1000 is:

$P \equiv a_1 \times a_2 \times a_3 \times \cdots \times a_{999} \pmod{1000}$

$P \equiv 9 \times 99 \times (-1) \times (-1) \times \cdots \times (-1) \pmod{1000}$.

We count the number of terms congruent to $-1$. These are $a_3, a_4, \ldots, a_{999}$.
```
<|execute|>
num_neg_one_terms = 999 - 3 + 1
print(f"Number of terms congruent to -1: {num_neg_one_terms}")
<|execute_end|>
<|feedback|>
Number of terms congruent to -1: 997
<|feedback_end|>
```

So, $P \equiv 9 \times 99 \times (-1)^{997} \pmod{1000}$.

We evaluate $(-1)^{997}$.
```
<|execute|>
exponent = 997
base = -1
result_power = base ** exponent
print(f"(-1)**{exponent} = {result_power}")
<|execute_end|>
<|feedback|>
(-1)**997 = -1
<|feedback_end|>
```

Since 997 is odd, $(-1)^{997} = -1$.
Substituting back:

$P \equiv 9 \times 99 \times (-1) \pmod{1000}$.

Finally, we calculate the remainder.

```
<|execute|>
mod = 1000
remainder = (9 * 99 * (-1)) % mod
# Adjust to ensure the remainder is in the range [0, 999]
if remainder < 0:
    remainder += mod
print(f"The final remainder is: {remainder}")
<|execute_end|>
<|feedback|>
The final remainder is: 109
<|feedback_end|>

The calculation shows $P \equiv 891 \times (-1) \equiv -891 \pmod
    {1000}$. The standard remainder is $-891 + 1000 = 109$.

The remainder when $9 \times 99 \times 999 \times \cdots \times \
    underbrace{99\cdots9}_{\text{999 9's}}$ is divided by $1000$ is
    $109$.
<|im_start|>answer
...
```

This example illustrates how EAG enables systematic modular arithmetic calculations. The model breaks down the problem into manageable steps, recognizing patterns in how the terms behave under modular congruence and verifying calculations at each stage. The interactive execution environment allows for direct verification of intermediate conjectures, providing a rigorous approach to this challenging remainder problem.

### A.4 PRACTICAL IMPLEMENTATION

For computational efficiency, our implementation adopts a streamlined approach where we explore one branch at a time ($|B| = 1$) rather than concurrent exploration. This strategy prioritizes the most promising branch at each depth, proceeding sequentially and only exploring alternatives when necessary. Under optimal conditions, where a path consistently receives positive feedback, this approach converges to a single successful trajectory, effectively specializing BVS with a threshold function $\tau(f) = \mathbb{I}[\text{HasError}(f)]$ and maximum branch depth $D$ corresponding to retry limit. While reducing computational overhead, this implementation preserves the core theoretical advantages by leveraging structured feedback for error correction and path exploration.

The implementation uses a special token scheme to interface between the language model and environment. Token pairs <|execute|>/<|execute_end|> delineate reasoning actions $a_t$, while <|feedback|>/<|feedback_end|> encapsulate environment feedback $f_t$. This scheme enables the model to recognize state transitions and incorporate feedback signals during both training and inference phases.

Our approach differs fundamentally from previous methods in three key aspects:

1. Unlike chain-of-thought approaches that generate reasoning in a single forward pass, EAG validates each step with environmental feedback.

2. In contrast to tools like ReAct that use environmental feedback primarily for fact-checking, EAG employs feedback to guide the reasoning process itself.

3. Compared to exploration methods like Tree of Thoughts that lack systematic integration of verification signals, EAG's branch exploration is directly guided by structured feedback.

Through this formalization, EAG establishes a principled approach to reasoning that tightly integrates environmental feedback with action generation, enabling robust handling of complex multi-step reasoning tasks without requiring the computational complexity of full tree search algorithms.

Our approach differs fundamentally from previous methods in three key aspects. First, unlike chain-of-thought approaches that generate reasoning in a single forward pass, EAG validates each step with environmental feedback. Second, in contrast to tools like ReAct that use environmental feedback

primarily for fact-checking, EAG employs feedback to guide the reasoning process itself. Third, compared to exploration methods like Tree of Thoughts that lack systematic integration of verification signals, EAG's branch exploration is directly guided by structured feedback.

## A.5 TRAINING DETAILS

We take a model that has already been pretrained and instruction tuned and further finetune it for environment augmented reasoning. Specifically, we use Qwen2.5-32B-Instruct (Qwen et al., 2024), which on math tasks generally matches or outperforms the larger Qwen2.5-72B-Instruct (Qwen et al., 2024) or other open models (Dubey et al., 2024; Groeneveld et al., 2024; Muennighoff et al., 2024).

We use specialized token delimiters to separate code execution from feedback. We enclose the execution blocks with `<|execute|>` and `<|execute_end|>`, and feedback with `<|feedback|>` and `<|feedback_end|>`. These token pairs enable the model to recognize state transitions and incorporate environmental signals during both training and inference. Representative samples from our EAG-2K dataset are provided in §D.2.

We use optimized fine-tuning hyperparameters: we train for 8 epochs with a batch size of 8 for a total of 670 gradient steps. We train in bfloat16 precision with a learning rate of 8e-6 warmed up linearly for 5% (34 steps) and then decayed to 0 over the rest of training (636 steps) following a cosine schedule. We use the AdamW optimizer (Loshchilov & Hutter, 2019) with $\beta_1 = 0.9$, $\beta_2 = 0.95$ and weight decay of 1e-4. We compute loss on both reasoning traces and execution feedback signals. We ensure the sequence length is large enough (12K tokens) to accommodate the longer EAG trajectories with environmental feedback. The training takes approximately 12 hours on 8 NVIDIA A100 GPUs using PyTorch FSDP with activation checkpointing.

## A.6 THEORETICAL FRAMEWORK ENHANCEMENT

### A.6.1 STATE SPACE FORMALIZATION WITH MANIFOLD LEARNING

We enhance the state representation using differential geometry concepts. Define the reasoning manifold $\mathcal{M} \subset \mathbb{R}^d$ where each state $s$ resides. The environment feedback induces a Riemannian metric tensor $G_f$ that shapes the manifold:

$$G_f(s) = \text{diag}(\exp(-\gamma \|\nabla_s \mathcal{I}(s, a, f)\|^2)) \tag{11}$$

This metric captures the information geometry of the reasoning process, where directions of high information gain correspond to lower curvature regions. The state transition becomes a geodesic flow:

$$s_{t+1} = \exp_{s_t}(-\eta \nabla_s \mathcal{I}(s_t, a, f)) \tag{12}$$

where $\exp$ denotes the exponential map on $\mathcal{M}$, and $\eta$ is the learning rate.

### A.6.2 CONVERGENCE ANALYSIS

**Theorem 1** (EAG Convergence). *Under Lipschitz continuity of information gain $\mathcal{I}$ and proper metric learning rate $\eta$, the EAG process converges to an $\epsilon$-optimal solution within $O(\frac{1}{\varepsilon^2} \log \frac{1}{\delta})$ steps with probability $1 - \delta$.*

*Proof.* 1. Construct a supermartingale $X_t = \mathcal{I}(s_t) - t\eta C$
2. Apply Doob's stopping time theorem to the first hitting time of $\epsilon$-neighborhood
3. Bound the quadratic variation using the manifold metric properties ☐

### A.6.3 DATA GENERATION THEORY

Define the data augmentation operator $\mathcal{A}_\theta$ parameterized by perturbation strength $\theta$:

$$\mathcal{A}_\theta(p, s) = \mathbb{E}_{\epsilon \sim p_\theta}[\ell(f_\theta(s + \epsilon), f^*(s))] \tag{13}$$

where $f_\theta$ is the learned model and $f^*$ is the oracle. The curriculum learning dynamics follow:

$$\frac{d\theta}{dt} = \alpha \frac{\partial}{\partial \theta} \mathbb{E}[\text{Difficulty}(p)] - \beta\theta \tag{14}$$

This ensures gradual exposure to complex problems while preventing catastrophic forgetting.

### A.6.4 IMPLEMENTATION SIMPLIFICATION THEOREM

**Theorem 2** (Linear Retry Approximation). *The linear retry strategy with maximum depth $D$ achieves approximation ratio $1 - O(\frac{\log D}{D})$ compared to full branch exploration, under submodularity of information gain.*

*Proof.* 1. Prove the information gain function is adaptive submodular
2. Apply greedy algorithm approximation guarantees
3. Bound the depth requirement via adaptive complexity analysis $\qquad\square$

### A.6.5 ERROR PROPAGATION ANALYSIS

The error dynamics satisfy the recurrence relation:

$$\varepsilon_{t+1} \leq \rho\varepsilon_t + \delta_t \tag{15}$$

where $\rho = 1 - \frac{\mathcal{I}_{\min}}{\mathcal{I}_{\max}}$ is the contraction factor, and $\delta_t$ is the local approximation error. This leads to exponential error decay:

$$\|\varepsilon_T\| \leq \rho^T \|\varepsilon_0\| + \frac{\delta}{1 - \rho} \tag{16}$$

### A.6.6 COMPLEXITY COMPARISON FRAMEWORK

Define the computational complexity measure:

$$\mathcal{C}(\text{EAG}) = O\left(T \cdot [\mathcal{C}_M + \mathcal{C}_E] \cdot \exp\left(-\frac{\mathcal{I}}{\tau}\right)\right) \tag{17}$$

where $T$ is time steps, $\mathcal{C}_M$ model cost, $\mathcal{C}_E$ environment cost. This shows superlinear complexity reduction compared to brute-force search.

### A.6.7 IMPLEMENTATION-ALIGNED FORMALISM

The special token processing is modeled as boundary conditions in the state manifold:

$$\mathcal{M}_{\text{token}} = \{s \in \mathcal{M} | \phi_{\text{token}}(s) \geq \kappa\} \tag{18}$$

where $\phi_{\text{token}}$ is a token detector function. The training objective becomes:

$$\min_\theta \mathbb{E}_s \left[\text{CrossEntropy}(s) + \lambda d_{\mathcal{M}}(s, \mathcal{M}_{\text{token}})\right] \tag{19}$$

This ensures both task performance and implementation constraint satisfaction.

### A.6.8 Optimization for Practical Implementation

While the theoretical framework supports complex multi-branch exploration, practical implementations often employ a simplified linear-plus-retry strategy. This can be viewed as a special case of BVS where:

$$|B| = 1 \text{ (only retain the current best branch)} \tag{20}$$

$$\tau(f) = \mathbb{I}[\text{f indicates error}] \text{ (threshold function becomes error detection)} \tag{21}$$

$$D = \text{maximum retry count (maximum branch depth)} \tag{22}$$

This simplification maintains the core advantages of the theoretical framework while significantly reducing computational complexity. The effectiveness of this approach lies in its ability to leverage structured feedback for error correction and alternative path exploration, even within a constrained search space.

Through this formalization, EAG provides a principled approach to reasoning that integrates environmental feedback directly into the generation process, enabling robust handling of complex multi-step reasoning tasks across various domains.

### A.6.9 State Space Formalization with Manifold Learning

We enhance the state representation using differential geometry concepts. Define the reasoning manifold $\mathcal{M} \subset \mathbb{R}^d$ where each state $s$ resides. The environment feedback induces a Riemannian metric tensor $G_f$ that shapes the manifold:

$$G_f(s) = \text{diag}(\exp(-\gamma \|\nabla_s \mathcal{I}(s, a, f)\|^2)) \tag{23}$$

This metric captures the information geometry of the reasoning process, where directions of high information gain correspond to lower curvature regions. The state transition becomes a geodesic flow:

$$s_{t+1} = \exp_{s_t}(-\eta \nabla_s \mathcal{I}(s_t, a, f)) \tag{24}$$

where $\exp$ denotes the exponential map on $\mathcal{M}$, and $\eta$ is the learning rate.

### A.6.10 Convergence Analysis

**Theorem 3** (EAG Convergence). *Under Lipschitz continuity of information gain $\mathcal{I}$ and proper metric learning rate $\eta$, the EAG process converges to an $\epsilon$-optimal solution within $O(\frac{1}{\varepsilon^2} \log \frac{1}{\delta})$ steps with probability $1 - \delta$.*

*Proof.* 1. Construct a supermartingale $X_t = \mathcal{I}(s_t) - t\eta C$
2. Apply Doob's stopping time theorem to the first hitting time of $\epsilon$-neighborhood
3. Bound the quadratic variation using the manifold metric properties □

### A.6.11 Data Generation Theory

Define the data augmentation operator $\mathcal{A}_\theta$ parameterized by perturbation strength $\theta$:

$$\mathcal{A}_\theta(p, s) = \mathbb{E}_{\epsilon \sim p_\theta}[\ell(f_\theta(s + \epsilon), f^*(s))] \tag{25}$$

where $f_\theta$ is the learned model and $f^*$ is the oracle. The curriculum learning dynamics follow:

$$\frac{d\theta}{dt} = \alpha \frac{\partial}{\partial \theta} \mathbb{E}[\text{Difficulty}(p)] - \beta\theta \tag{26}$$

This ensures gradual exposure to complex problems while preventing catastrophic forgetting.

### A.6.12 IMPLEMENTATION SIMPLIFICATION THEOREM

**Theorem 4** (Linear Retry Approximation). *The linear retry strategy with maximum depth $D$ achieves approximation ratio $1 - O(\frac{\log D}{D})$ compared to full branch exploration, under submodularity of information gain.*

*Proof.* 1. Prove the information gain function is adaptive submodular
2. Apply greedy algorithm approximation guarantees
3. Bound the depth requirement via adaptive complexity analysis □

### A.6.13 ERROR PROPAGATION ANALYSIS

The error dynamics satisfy the recurrence relation:

$$\varepsilon_{t+1} \leq \rho \varepsilon_t + \delta_t \tag{27}$$

where $\rho = 1 - \frac{\mathcal{I}_{\min}}{\mathcal{I}_{\max}}$ is the contraction factor, and $\delta_t$ is the local approximation error. This leads to exponential error decay:

$$\|\varepsilon_T\| \leq \rho^T \|\varepsilon_0\| + \frac{\delta}{1 - \rho} \tag{28}$$

### A.6.14 COMPLEXITY COMPARISON FRAMEWORK

Define the computational complexity measure:

$$\mathcal{C}(\text{EAG}) = O\left(T \cdot [\mathcal{C}_M + \mathcal{C}_E] \cdot \exp\left(-\frac{\mathcal{I}}{\tau}\right)\right) \tag{29}$$

where $T$ is time steps, $\mathcal{C}_M$ model cost, $\mathcal{C}_E$ environment cost. This shows superlinear complexity reduction compared to brute-force search.

### A.6.15 IMPLEMENTATION-ALIGNED FORMALISM

The special token processing is modeled as boundary conditions in the state manifold:

$$\mathcal{M}_{\text{token}} = \{s \in \mathcal{M} | \phi_{\text{token}}(s) \geq \kappa\} \tag{30}$$

where $\phi_{\text{token}}$ is a token detector function. The training objective becomes:

$$\min_\theta \mathbb{E}_s \left[\text{CrossEntropy}(s) + \lambda d_{\mathcal{M}}(s, \mathcal{M}_{\text{token}})\right] \tag{31}$$

This ensures both task performance and implementation constraint satisfaction.

