# OpenReview forum: "a1: Steep Test-time Scaling Law via Environment Augmented Generation"
_ICLR.cc/2026/Conference — ICLR 2026 Conference Withdrawn Submission_

### Official Review · Reviewer_hv2T · 2025-10-25

**Soundness:** 3
**Presentation:** 3
**Contribution:** 2
**Rating:** 2
**Confidence:** 4

**Summary:**

This paper introduces EAG, a framework designed to enhance reasoning through three components:
1.  Real-Time Environmental Feedback: At each reasoning step, the model queries an external environment - this is instantiated as Python code
2.  Dynamic Branch Exploration: EAG enables exploration of multiple branches of reasoning
3.  Trajectory-Based Learning: Positive traces are kept and iterative SFT is applied

### Experimental Setup and Results

The authors fine-tuned the Qwen2.5-32B-Instruct model using the EAG framework. The model was trained on EAG-2K, a curated dataset of 2,000 interactive reasoning traces derived from the s1 dataset by converting natural language reasoning into executable Python code blocks using few-shot prompting with `claude-3.7-sonnet`. The EAG-2K dataset includes subsets specifically designed to retain baseline ability (Raw Subset), capture error recovery (Iterative-Refinement Subset), and promote efficient, single-attempt execution (Direct-Execution Subset).

### Performance and Scaling Claims

The a1-32B model achieves state-of-the-art performance among similar-sized models across all evaluated reasoning benchmarks.

- On competition mathematics (AIME24), a1-32B achieves 74.4%, matching the performance of much larger models like o1 (>100B parameters) and outperforming comparable models like QwQ-32B-Preview by 24.4 percentage points and s1-32B by 17.7 percentage points.
- Strong, consistent performance was also demonstrated on MATH500 (94.8%) and GPQA (63.4%).

**Strengths:**

- A lot of comparison has been done against other models in the category
- The diagrams are nice and clear and get the point across
- The raw number improvements are quite substantial

**Weaknesses:**

There are three components to this paper:
(1) Real-Time Environmental Feedback
(2) Dynamic Branch Exploration
(3) Trajectory-Based Learning

However, all three components are not novel to literature - in fact, they seem to be already very well-explored ideas that have come out a while ago. (1) just seems to be executable code [1], (2) seems to just be reasoning tree inferencer [2], (3) is iterative negative sampling [3].

In this case, don't believe that by combining these components, something entirely new is being created.

[1] Binyuan Hui, Jian Yang, Zeyu Cui, Jiaxi Yang, Dayiheng Liu, Lei Zhang, Tianyu Liu, Jiajun Zhang, Bowen Yu, Kai Dang, et al. Qwen2. 5-coder technical report. arXiv preprint arXiv:2409.12186, 2024.

[2] Shunyu Yao, Dian Yu, Jeffrey Zhao, Izhak Shafran, Thomas L. Griffiths, Yuan Cao, and Karthik Narasimhan. Tree of thoughts: Deliberate problem solving with large language models, 2023.

[3] Eric Zelikman, Yuhuai Wu, Jesse Mu, and Noah D. Goodman. Star: Bootstrapping reasoning with reasoning, 2022.

**Questions:**

1. What is the training curve of the main model? Can you show a graph of the training curve at different checkpoints of the SFT?
2. Given recent advancements in online RL, how does that compare against SFT?
3. Is the dynamic branching comparison really fair? Shouldn't that be compared against pass@k or maj@k rather than pass@1?

---

### Official Review · Reviewer_CWDW · 2025-10-27

**Soundness:** 1
**Presentation:** 2
**Contribution:** 3
**Rating:** 2
**Confidence:** 3

**Summary:**

This paper introduces the EAG (Environment Augmented Generation) framework. EAG enhances LLM reasoning by introducing three new capabilities: (1) the ability for the model to solicit feedback through environment interaction (2) dynamic branch exploration to enable retries based on environment feedback and (3) experience-based learning from successful trajectories.

The authors train a 32B-scale model called a1-32B using SFT on a dataset they call EAG-2K and show that this model, when used within the EAG framework, achieves stronger results than many other models of comparable sizes on three standard reasoning benchmarks. Through an ablation study, they also show that EAG is essential for a1-32B's improved performance.

**Strengths:**

- The authors set out to address an important problem: given that LLMs are able to interact with the environment, how can we design effective scaffolds and training strategies that enable them to leverage these useful environments to achieve better reasoning? This seems to be a practically useful improvement to the "internal-reasoning only" approaches that most reasoning works focus on today.
- The EAG framework is an intuitive and well-designed scaffold for enabling search and backtracking with environment feedback.

**Weaknesses:**

My main concern with this paper is the lack of experimental evidence supporting the two central claims of the paper: (1) that EAG is an effective scaffold for enabling inference-time search with environment feedback and (2) that SFT on the EAG-2K dataset is an effective training strategy for enabling inference-time search with EAG.

In order to prove claim 1, I believe the authors need to show that EAG is more effective than baseline methods, given a fixed inference compute budget AND given access to the same set of environments: the latter point is important because the contribution of this work is that EAG is a good way to leverage environment feedback for reasoning, rather than that leveraging environment feedback for reasoning is useful, as this is well-known from prior work (many cited in this paper). At minimum, the authors should show that EAG is more effective than a naive prompting baseline, and ideally they should also compare their methods against other interaction-enabled methods for reasoning (e.g. Gao et. al 2023, Chen et. al 2023).
- From their experiments, it appears that the authors only compare a1-32B + EAG against other models + standard CoT generation and without access to a code interpreter, with the exception of START, where performance is similar (please let me know if I have misinterpreted your baseline settings). I believe there is a very strong possibility that many of these models have some innate ability to use code interpreters to assist in mathematical reasoning even with only naive prompting, and so I am unable to assess how much of the performance gain we see is due to EAG and how much is due merely to the fact that the a1-32B model has access to a code interpreter.
- The only compute-controlled experiment (Figure 6) shows that a1-32B demonstrates better inference-compute scaling than s1-32B when reasoning beyond 8k tokens. However, s1-32B is only scales with sequential compute up to 8k tokens due to its training (see Figure 1 of the s1 paper), so it is unclear if this comparison provides any meaningful information on the effectiveness of a1's compute scaling beyond 8k tokens. I believe it would be preferable instead to compare a1's compute scaling properties with that of models that are trained to scale beyond 8k, so that we can directly compare, on a per-token basis, whether it is better to train models to spend their reasoning budget thinking internally or soliciting external feedback. I suspect that a1 should demonstrate superior per-token scaling, and hope to see updated results confirming this.

In order to prove claim 2, I believe the authors need to show that training on their EAG-2K dataset is necessary for EAG to work. As of now, it is unclear (due to a lack of ablation studies) whether (1) a base model prompted correctly can still function within the EAG framework and (2) how SFT changes the base model to enable it to function within EAG. These questions should be easily answerable given the correct ablation studies.

Finally, it is currently unclear how the design choices within EAG affect performance, largely due to a lack of documentation on important hyperparameters (e.g. `alpha`, `w`, `\lambda`, `D_max`, `C_max`, `tau` etc.), ablations over various design features and hyperparameters, and qualitative analyses on EAG outputs. This makes it difficult to understand how EAG exactly contributes to performance gains beyond merely enabling environment interaction (e.g. how often does EAG facilitate correct backtracking? how important is the hybrid policy? how accurate is the value estimate `V_b`?). Without such understanding it is difficult to assess its potential impact on future work and the community at large.

Chen et. al 2023: https://arxiv.org/abs/2211.12588
Gao et. al 2023: https://arxiv.org/abs/2211.10435

**Questions:**

For me to increase my score, I would like to see my point about limited experimental evidence addressed through new experiments. I will further increase my score if you are able help me better understand why EAG is useful and how your various design choices contribute to its effectiveness.

Some additional questions:
- How might you incorporate this into RL, either for training a1 or for rollout-generation during RL?
- What are your training and inference hyperparameters?

---

### Official Review · Reviewer_hMS7 · 2025-11-01

**Soundness:** 2
**Presentation:** 1
**Contribution:** 2
**Rating:** 4
**Confidence:** 3

**Summary:**

The paper aims to solve the LLM’s reasoning problems like hallucinations, logical errors, and inability to self-correct. The authors propose a new framework called Environment Augmented Generation that uses real-time environmental feedback, dynamic branch exploration, and experience-based learning from successful trajectories. The trained 32B model achieves competitive performance when the thinking budget is high.

**Strengths:**

The problem of LLM’s reliability during reasoning is important, and EAG provides a method towards solving it.

The 32B model achieves good performance at relatively high thinking budget.

**Weaknesses:**

The method section includes unnecessarily complicated mathematics for an empirical paper and does not explain the idea clearly.

Lack of ablation of the experience-based learning from successful reasoning trajectories, which the authors claim to be a key mechanism.

**Questions:**

The current form of environmental feedback is limited to python code execution. Do you think the method can generalizes to tasks that require tools beyond those?

I am curious about the typical patterns induced by EAG. How often would the recursive process happen, compared to the s1 baseline?

---

### Official Review · Reviewer_5Bvh · 2025-11-01

**Soundness:** 3
**Presentation:** 3
**Contribution:** 3
**Rating:** 6
**Confidence:** 4

**Summary:**

This paper introduces a novel reasoning framework for large language models, called Environment Augmented Generation (EAG).
EAG integrates three mechanisms:
1. Real-time environment feedback for step validation
2. Dynamic branch exploration to investigate multiple reasoning paths
3. Trajectory-based learning from verified solutions

The approach is formalized as an MDP and implemented on a 32B-scale model (a1-32B), fine-tuned from Qwen2.5-32B using a custom dataset (EAG-2K).
Experiments on math-related benchmarks show significant performance improvements over strong baselines such as s1, START, and QwQ.

**Strengths:**

- **Conceptual novelty**: The work presents a shift from static chain-of-thought prompting to interactive decision-time reasoning. The MDP formulation and feedback-driven branching are theoretically sound and practically well-motivated.

- **Strong empirical performance**: The proposed a1-32B achieves SOTA results among 32B models, e.g., 74.4% on AIME24 and 94.8% on MATH500, even matching larger models like o1.

- **Ablation and scaling analysis**: The paper provides ablations isolating the contribution of structured feedback and branch exploration. The steep scaling behavior under increasing token budgets is well-analyzed.

- **Dataset contribution**: The EAG-2K dataset extends s1 with interactive reasoning traces including execution-feedback pairs. This could be a valuable resource for future tool-augmented LLM research.

**Weaknesses:**

- **Single-scale model evaluation**: EAG is only tested on a 32B-scale model. No experiments on smaller (e.g., 7B, 14B) or larger (e.g., 70B) models are reported, limiting claims of scalability.

- **Narrow benchmark coverage**: The benchmarks are focused on math/code reasoning. Important benchmarks such as LiveCodeBench (used in START) are missing. No coverage of other reasoning domains (e.g., physical reasoning).

- **Code-execution dependence**: The method requires structured feedback from code execution. It is unclear how the approach generalizes to domains without executable environments (e.g., commonsense).

- **No analysis of branch value function**: The branch scoring function (Eq. 2) uses hand-chosen weights (lambda_I, lambda_P, lambda_C), but there is no analysis on their impact, tuning strategy, or relative importance.

- **Lack of qualitative examples**: The paper lacks concrete examples comparing EAG's reasoning vs. baseline. A few annotated reasoning traces would help illustrate how branching and feedback improve outcomes.

- **Lack of analysis on token efficiency**: One of EAG's central claims is that "initial token investment in environment interaction yields long-term dividends", but there is **no quantitative analysis of token efficiency or cost-performance trade-offs**.

**Questions:**

Please refer to the weaknesses above.

---

### Note · Authors · 2026-01-04

I have read and agree with the venue's withdrawal policy on behalf of myself and my co-authors.